# Growth Hormone Improves Adipose Tissue Browning and Muscle Wasting in Mice with Chronic Kidney Disease-Associated Cachexia

**DOI:** 10.3390/ijms232315310

**Published:** 2022-12-04

**Authors:** Robert H. Mak, Sujana Gunta, Eduardo A. Oliveira, Wai W. Cheung

**Affiliations:** 1Division of Pediatric Nephrology, Rady Children’s Hospital, University of California, San Diego, CA 92093, USA; 2Pediatric Services, Vista Community Clinic, Vista, CA 92084, USA; 3Department of Pediatrics, Health Sciences Postgraduate Program, School of Medicine, Federal University of Minas Gerais (UFMG), Belo Horizonte 30310-100, Brazil

**Keywords:** chronic kidney disease, growth hormone, cachexia, lipolysis, adipose tissue browning, muscle mass, muscle function

## Abstract

Cachexia associated with chronic kidney disease (CKD) has been linked to GH resistance. In CKD, GH treatment enhances muscular performance. We investigated the impact of GH on cachexia brought on by CKD. CKD was induced by 5/6 nephrectomy in c57BL/6J mice. After receiving GH (10 mg/kg/day) or saline treatment for six weeks, CKD mice were compared to sham-operated controls. GH normalized metabolic rate, increased food intake and weight growth, and improved in vivo muscular function (rotarod and grip strength) in CKD mice. GH decreased uncoupling proteins (UCP)s and increased muscle and adipose tissue ATP content in CKD mice. GH decreased lipolysis of adipose tissue by attenuating expression and protein content of adipose triglyceride lipase and protein content of phosphorylated hormone-sensitive lipase in CKD mice. GH reversed the increased expression of beige adipocyte markers (UCP-1, CD137, Tmem26, Tbx1, Prdm16, Pgc1α, and Cidea) and molecules implicated in adipose tissue browning (Cox2/Pgf2α, Tlr2, Myd88, and Traf6) in CKD mice. Additionally, GH normalized the molecular markers of processes connected to muscle wasting in CKD, such as myogenesis and muscle regeneration. By using RNAseq, we previously determined the top 12 skeletal muscle genes differentially expressed between mice with CKD and control animals. These 12 genes’ aberrant expression has been linked to increased muscle thermogenesis, fibrosis, and poor muscle and neuron regeneration. In this study, we demonstrated that GH restored 7 of the top 12 differentially elevated muscle genes in CKD mice. In conclusion, GH might be an effective treatment for muscular atrophy and browning of adipose tissue in CKD-related cachexia.

## 1. Introduction

Cachexia in chronic kidney disease (CKD) results in profound loss of adipose tissue and muscle mass [1,2]. Although poor protein-calorie intake is a major factor, growth hormone (GH) resistance has been linked to CKD-associated cachexia [1,2,3,4]. GH increases muscle strength in healthy men [5]. Short-term administration of recombinant GH increases muscle protein synthesis and muscle mass as well as improves quality of life in hemodialysis patients [6,7,8,9,10]. However, most studies have used dual energy X-ray absorptiometry (DXA) to measure muscle mass as a surrogate marker for the effect of GH treatment in hemodialysis patients, but the validity of this extrapolation in CKD is questionable since DXA cannot differentiate between a true increase in muscle mass versus fluid overload [11]. Moreover, the effect of GH on muscle function in CKD has not been adequately studied [12]. We have previously described the pathways involved in muscle wasting in a mouse model of CKD [13]. IGF-I and myostatin represent yin-and-yang signaling pathways in the pathogenesis of CKD-associated cachexia muscle wasting [14]. GH resistance in CKD, due to signal transduction defects in JAK-STAT pathways, is associated with upregulated SOCS-2 and downregulated IGF-I in skeletal muscle [15]. Myostatin is overexpressed in CKD-associated wasting and is accompanied by increased protein degradation via FoxOs, Atrogin-1, and MuRF-1, and decreased myogenesis via Pax3 and MyoD [13]. 

Adipose tissue regulates whole-body energy metabolism. White adipose tissue (WAT) is a key energy reservoir, while brown adipose tissue (BAT) is involved in the regulation of thermogenesis [16]. Recent studies have demonstrated that WAT browning, a process characterized by a phenotypic transition from WAT to thermogenic BAT, is implicated in the pathogenesis of cachexia. Indeed, browning of WAT preceded skeletal muscle atrophy in mouse models of CKD and cancer [17,18]. GH regulates adipose tissue metabolism [19,20]. In this study, we investigate the effects and mechanisms of GH in a mouse model of CKD, with emphasis on adipose tissue browning and muscle wasting.

## 2. Results

### 2.1. GH Stimulates Food Intake and Increases Body Weight in CKD Mice 

We empirically determined the optimal dose of GH treatments in our mouse model of CKD. Six-week-old c57BL/6J male mice were used for this study. Schematic representation of the experimental design is shown in Figure 1A. CKD in mice was induced by a two-stage subtotal nephrectomy, while a sham procedure was performed in control mice [13]. GH treatment was initiated in eight-week-old CKD or sham mice. CKD or sham mice were treated with recombinant human GH (5 mg/kg/day or 10 mg/kg/day, intraperitoneal) or vehicle for six weeks. During the treatment, all mice were housed in individual cage and fed ad libitum. Dietary intake as well as weight gain for each mouse was recorded weekly. Mice were sacrificed at the age of 14 weeks old. Serum and blood chemistry of CKD and sham mice are listed (Table 1). CKD mice were uremic, as CKD mice had a higher concentration of BUN and serum creatinine than control mice. Over the course of the six-week ad libitum experiment, GH stimulated food intake and improved weight gain in both CKD and sham mice. GH-treated CKD and GH-treated sham mice exhibited significantly more average daily energy intake and weight gain compared to vehicle-treated CKD and vehicle-treated sham mice, respectively (Figure 1B,C). More importantly, we found that CKD mice treated with 10 mg/kg/day demonstrated significantly improved food intake and weight gain relative to CKD mice treated with 5 mg/kg/day or vehicle. As a result, daily dosing of 10 mg/kg of GH for CKD mice was selected for the subsequent food-restrictive study. 

### 2.2. GH Improves Energy Homeostasis in CKD Mice

We utilized a food-restrictive strategy to study the pharmacological effects of GH in CKD mice beyond appetite stimulation and their consequent body weight gain (Figure 1D). Two-stage subtotal nephrectomy for CKD mice and a sham procedure for control mice were also performed. Eight-week-old CKD or sham mice were housed individually. Mice were given GH (10 mg/kg/day, intraperitoneal) or vehicle for six weeks. For this diet-restrictive study, vehicle-treated CKD mice were fed ad libitum, while the other mouse groups (GH-treated CKD mice as well as GH-treated or vehicle-treated sham mice) received an energy intake amount equal to that of vehicle-treated CKD mice (Figure 1E). Mice were sacrificed at the age of 14 weeks old. Serum and blood chemistry of mice are listed in Table 2. Vehicle- or GH-treated CKD mice were uremic, as they had a higher concentration of BUN and serum creatinine than sham mice. We verified that daily GH treatments resulted in high circulating concentrations of human GH in mice. Mean circulating human GH was not different between GH-treated CKD (325.3 ± 65.3 µg/L) and GH-treated control mice (364.6 ± 76.4 µg/L), whereas no human GH was detected in CKD or control mice receiving vehicle. A significant increase in weight gain in GH-treated CKD mice relative to vehicle-treated CKD mice was observed at day 21, and the trend remained significant for the rest of the study (Figure 1F). In addition, GH normalized fat and lean mass content, weight of gastrocnemius, resting metabolic rate, and in vivo muscle function (rotarod activity and grip strength) in CKD mice (Figure 1G–L).

### 2.3. GH Improves Skeletal Muscle and Adipose Tissue Energy Homeostasis in CKD Mice

For the rest of the investigation, gastrocnemius, WAT, and BAT tissue from the diet-restrictive study were used. We studied the effects of GH on skeletal muscle and adipose tissue energy homeostasis in CKD mice. Protein content of UCPs in gastrocnemius as well as in WAT and BAT was significantly higher in vehicle-treated CKD mice (Figure 2A,C,E). Inversely, ATP content in gastrocnemius, WAT, and BAT was significantly lower in vehicle-treated CKD mice (Figure 2B,D,F). GH decreased UCPs but increased ATP content in muscle and adipose tissue in CKD mice.

### 2.4. GH Mitigates Lipolytic Enzymes in CKD Mice

Elevated lipolysis is important for adipose tissue wasting in cachexia [21]. We investigated the molecular basis for the loss of adipose tissue in CKD mice. Inguinal WAT gene expression and protein content of adipose triglyceride lipase (ATGL) was significantly increased in vehicle-treated CKD mice (Figure 3A,B). Inguinal WAT gene expression and protein content of hormone-sensitive lipase (HSL) was not different among groups of mice (Figure 3C,D). However, phosphorylated HSL Ser552 protein content in inguinal WAT, a surrogate marker for protein kinase A-activated lipolysis, was five-fold higher in vehicle-treated CKD mice compared to control mice (Figure 3E). Importantly, GH significantly decreased inguinal WAT gene expression and protein content of ATGL as well as protein content of phosphorylated HSL in CKD mice.

### 2.5. GH Mitigates White Adipose Tissue Browning in CKD Mice

Beige adipocyte cell surface markers’ (CD137, Tbx1, Tmem26, Prdm16, Pgc1α, and Cidea) mRNA expression in inguinal WAT was normalized or decreased in GH-treated CKD mice relative to vehicle-treated CKD mice (Figure 4A–F). In WAT, de novo browning recruitment is promoted by the activation of Cox2/Pgf2α pathway and toll-like receptor Tlr2 and adaptor molecules, such as Myd88 and Traf6 [22]. GH treatment normalized expression of inguinal WAT Cox2, Pgf2α, Tlr2, Myd88, and Traf6 in CKD mice (Figure 4G–K).

### 2.6. GH Attenuates Muscle-Wasting Signaling and GH Resistance Pathways in CKD Mice

Perturbations of metabolic pathways lead to skeletal muscle atrophy in the cachexia and sarcopenia. Proinflammatory cytokines induce the catabolic pathways in muscle [1,2]. Treatment of GH attenuated gastrocnemius mRNA expression of inflammatory cytokines (Il1β, Il6, and Tnfα) in CKD mice (Figure 5A–C). GH ameliorated muscle regeneration and myogenesis by decreasing the mRNA expression of negative regulators of skeletal muscle mass (Atrogin-1, Murf-1, Myostatin, and Soc2) while increasing the mRNA expression of promyogenic factors (MyoD, Myogenin, Pax-7, and IGF-I) in CKD mice (Figure 5D–G). In agreement with previous observations [14,15], we also found impaired JAK2/STAT5 signaling in gastrocnemius muscle in CKD mice (Figure 5L,M). GH normalized muscle protein content of phosphorylated JAK2 and STAT5 in CKD mice.

### 2.7. Molecular Mechanism of GH on Muscle Function by RNAseq Analysis

We previously performed transcriptomic profiling of muscle wasting in CKD by RNAseq analysis and identified 12 differentially expressed genes in muscle [23]. Perturbations of these 12 muscle genes are correlated with impaired muscle and neuron regeneration, enhanced muscle thermogenesis, and fibrosis. Hence, we studied the effects of GH on expression of these 12 muscle genes in CKD mice. Notably, GH normalized or attenuated 7 out of those 12 differentially expressed muscle genes identified in CKD mice, while the expression of 5 muscle genes remained different in GH-treated CKD mice (Figure 6A–L).

## 3. Discussion

Patients with CKD frequently have cachexia, which has been linked to higher morbidity and mortality rates [1]. We looked into how GH affected cachexia in CKD mice. First, we showed that intraperitoneal administration of GH significantly increased caloric intake and weight growth in CKD mice (Figure 1B,C). We also demonstrated that the beneficial metabolic benefits of GH go beyond appetite stimulation. GH improves organismal metabolism (Figure 1F–L) as well as specific tissue energy balance (skeletal muscle and adipose tissue) in CKD mice (Figure 2). The findings of this study are consistent with those of other, earlier studies. In hemodialysis patients, GH enhances nutrition intake and increases lean body mass [8,9,10]. With the help of GH, the body’s anabolism is stimulated, and protein accretion happens in the muscles and extramuscular tissues [6,7,24].

By increasing muscle mass and improving energy efficiency, GH may improve muscle strength. Anaerobic and aerobic energy sources comprise the continuum of energy needed to fuel muscular function. Anaerobic energy systems are stimulated by GH, which suppresses the aerobic energy system. This increases muscle strength. After six months of GH therapy, healthy males showed a considerable improvement in their lower body muscle strength [5]. By “uncoupling” ATP synthesis, UCPs regulate energy homeostasis by dissipating the mitochondrial proton gradient for ATP synthesis and producing heat [25,26]. UCP3 is expressed in skeletal muscle, and upregulation of UCP3 has been reported in various conditions characterized by skeletal muscle atrophy, including denervation, diabetes, cancer, and sepsis [27]. Gastrocnemius UCP3 protein content along with ATP content was normalized in GH-treated CKD mice (Figure 2A,B). Putative functions of UCP3 are controversial. Interesting evidence for and against UCP3 involvement in thermogenesis has been published [27,28]. Furthermore, increased muscle expression of UCP3 has been postulated to modulate oxidative stress and lipotoxicity in a rat model of cachexic sepsis [29]. GH therapy reduced abnormal UCP1 and ATP content in WAT and BAT in CKD mice (Figure 2C–F). However, the precise role of UCP1 in disease-associated cachexia in humans is still a topic of debate. Several studies have described UCP1 expression, a biomarker of WAT browning, as a critical component of WAT dysfunction in cancer cachexia [17,18,30]. Results also suggested that a UCP-1 independent cascade could also regulate adipocyte homeostasis and influence tumor-induced WAT wasting [31]. Moreover, activation of BAT has been associated with hypermetabolism in cachexia, but information from human studies is scarce. A recent study investigated the relationship between activation of BAT and hypermetabolism in patients with emphysematous COPD (chronic obstructive pulmonary disease). BAT activity and gene expression of beige markers of BAT in WAT (Tmem26, Cidea, CD137, Shox2, and Tnfrsf9) were not different between COPD patients versus controls [32]. Medications may influence the sympathetic nervous system and BAT metabolism. Adrenergic receptor blockers and calcium channel blockers are commonly used by COPD patients. Involvement of β-adrenergic receptor signaling in BAT metabolism was reported in humans and rodents [33,34]. Data also indicated that calcium channel blockers regulated adipogenesis and BAT browning [35,36].

CKD-associated cachexia is a progressive, multifactorial metabolic syndrome that results in significant loss of adipose tissue and skeletal muscle mass. Fat loss from adipose tissue in CKD-associated cachexia may be due to the increased rate of lipolysis. Recent longitudinal studies found that the magnitude of adipose tissue wasting predicts poorer survival in cancer patients [37,38,39]. The bulk of lipid mobilization from adipose tissue is mediated through lipolysis. In canonical adipose tissue lipolysis, triglycerides stored in lipid droplets are hydrolyzed by ATGL and HSL to produce free glycerol and fatty acids and fuel peripheral tissue metabolism [40]. ATGL is the rate-limiting lipase and hydrolyzes triacylglycerol in lipid droplets to diacylglycerol. GH treatment attenuated inguinal WAT mRNA expression and protein content of ATGL in CKD mice (Figure 3A,B). Previous studies have shown increased ATGL expression in the adipose tissue of cancer-associated cachectic animals and humans [21,41,42]. Inguinal WAT gene expression and protein content of HSL was not different among groups of mice (Figure 3C,D). However, phosphorylated HSL Ser552 protein content in inguinal WAT, a surrogate marker for protein kinase A-activated lipolysis [43], was significantly increased in CKD mice (Figure 3E). Importantly, GH attenuated inguinal WAT protein content of phosphorylated HSL in CKD mice. Evidence of enhanced protein kinase A-activated lipolysis correlated with elevated whole-organism energy expenditure and increased adipose tissue thermogenesis, and increased expression of biomarkers of adipose tissue browning in WAT was reported in a mouse model of cancer cachexia [21]. Moreover, increased WAT protein content of phosphorylated HSL and protein Kinase A was also shown in a mouse model of CKD [44].

Browning of adipose tissue is associated with a hypermetabolic state and cachexia. Adipose tissue browning is evident in animal models of CKD-associated cachexia and cancer as well as in cachectic cancer patients [17,18,30]. We demonstrated that in CKD mice, GH reduced the browning of adipose tissue. The expression of biomarkers of beige adipocyte in WAT (CD137, Tbx-1, Tmem26, Prdm16, Pgc1a, and Cidea) was attenuated in CKD mice treated with GH (Figure 4A–F). Cox2/Pgf2 and inflammatory Tlr2, MyD88, and Traf6 signaling pathways have been associated with the biogenesis of browning [22]. GH treatment restored the expression of inflammatory molecules (Tlr2, MyD88, and Trap6) in the inguinal WAT of CKD mice treated with GH (Figure 4G–K). GH influences the metabolism of adipose tissue by binding to the GH receptor (GHR). Disrupted GH/GHR in mice results in multiple metabolic disorders. Global or adipose-specific GHR-deficient mice fail to demonstrate metabolic adaptability when challenged with a high-fat diet or cold temperature [45].

We looked at how GH affected the expression of molecules that control skeletal muscle metabolism in CKD mice. GH increases the expression of promyogenic factors (MyoD, Myogenin, and Pax-7) while decreasing or normalizing the expression of negative regulators of skeletal muscle mass (Atrogin-1, Murf-1, Myostatin, and Soc2, and inflammatory cytokines IL-1β, IL-6, and TNFα) (Figure 5A–J). Recent research indicates that the immune system and the GH/IGF-I axis interact in complicated and bidirectional ways. For example, the GH/IGF-I axis may be suppressed by inflammatory cytokines such as IL-1, IL-6, and TNFα, while GH/IGF-I may also influence systemic inflammation [46]. In cancer cachectic mice, IL-6 causes a decrease in fat content and stimulates adipose tissue browning [47]. In children with GH deficiency, GH has been found to reduce serum concentrations of IL-1β and TNFα [48]. In addition, GH lowers the serum concentrations of TNFα in adult hemodialysis patients [8]. Skeletal muscle growth and repair are influenced by the transcription factors Pax-3 and Pax-7. Pax-3 and Pax-7 regulate MyoD and myogenin [49]. MyoD and Myogenic Factor 5 (Myf5) are required to promote myogenic precursors. A downstream target of MyoD, myogenin controls the differentiation of myoblasts into myocytes and myotubes [49,50]. 

Because GH and IGF-I are powerful anabolic hormones that stimulate muscle mass increase and are crucial for maintaining skeletal mass, muscle loss in CKD has been linked to disruptions in the GH/IGF axis. As a result, IGF-I resistance may be a factor in the wasting of muscle in CKD [4]. In fact, GH therapy improved muscle mass compared to height in children with CKD [51]. Patients receiving continuous hemodialysis experienced an increase in blood IGF-I concentration following GH therapy [52,53,54]. The IGF-I signaling pathway, which promotes the proliferation and differentiation of satellite cells into myoblasts and the development of new myofibers, is one of the mechanisms by which GH affects skeletal muscle metabolism [55]. After a prolonged denervation injury, GH enhances muscle reinnervation, nerve regeneration, and functional outcomes [56]. Furthermore, Gautsch et al. have demonstrated that GH stimulates endocrine IGF-I-stimulated protein accretion, enhancing somatic and skeletal muscle growth in malnourished rats [57]. Interesting findings also point to a possible IGF-I-independent mechanism by which GH may exert anabolic effects in muscle [58]. Muscle wasting associated with CKD is brought on by an impaired JAK-STAT signal [14,15]. We have verified that recombinant human GH treatments resulted in high circulating concentration of human GH in CKD and control mice (Table 2). Muscle expression of IGF-I was decreased in CKD mice, and GH treatments normalized muscle IGF-I expression as well as restored the phosphorylated JAK2 and STAT5 muscle protein levels to normal in CKD mice (Figure 5K–M). Previous studies also showed that GH treatment increased muscle mRNA expression of IGF-I and attenuated JAK-STAT signaling in rodent models of CKD [14]. 

The GH dose administered to the mice in this study was about 200-fold higher than the dose typically used in humans. The recommended dose approved for treatment of growth failure in children with CKD is 0.35 mg/kg per week [59], whereas we used 10 mg/kg/day in mice for this study. However, our dose was comparable to those commonly used in rodent studies [60], and the observation of increased muscle mRNA expression of IGF-I as well as JAK/STAT phosphorylation after GH treatment in our CKD mice (Figure 5K–M) argues against any effect of GHR saturation. 

IGF-I is the most important downstream mediator of GH. Thus, IGF-I is generally considered to be the most important biomarker of GH action, as reflected in the inclusion of serum concentrations of IGF-I in the current guidelines for diagnosis and treatment of GH disorders in humans [61]. We have not measured serum IGF-I concentration in this study, as recent studies suggested that concentration of serum IGF-I is not a reliable marker for exogenous growth hormone activity in mice [60]. Male and female mice from four different strains of mice, including the 57BL/6J mouse strain used in this study, were treated with recombinant human GH (500 ug/day, intraperitoneally, for a period of 14 days). The total amount of GH administrated to mouse is ~7 mg in their study [60], which is comparable to the dose we used for the diet-restrictive study (total amount of GH is ~9.2 mg, presumably 22 g of body weight for CKD mice). In agreement to our observation (serum concentration of human GH in Table 2), GH treatment resulted in high circulating concentrations of human GH in all four strains of mice, whereas no human GH was detectable in control mice receiving isotonic 0.9% NaCl as vehicle. Two weeks of daily GH treatment significantly increased body and organ weight in male and female mice of all four inbred mouse strains when compared with controls. GH treatment failed to affect circulating (total) IGF-I concentrations in all strains and in both sexes. The liver is the main source of circulating IGF-I [62]. Hepatic expression of IGF-I mRNA did not show any difference between GH-treated mice versus control mice in any of these four strains of mice and sexes [60]. List et al. investigated the effects of GH in a mouse model of diet-induced diabetes [63]. Male c57BL/6J mice were fed a high-fat diet to induce obesity and type 2 diabetes. Subsequently, obese and diabetic mice were treated with various doses of GH for a period of six weeks. Comparable to our findings in CKD mice (Figure 1C,F,I), their highest dose of GH (215 μg/day/mouse for their study versus 220 μg/day/mouse in our study, presumable 22 g of body weight for CKD mice) resulted in a significant increment of total body mass and lean mass content. However, in contrast to the findings that the treatment of GH did not influence serum concentration of IGF-I in mice [60], GH treatment led to a significant increase in serum IGF-I in diabetic mice [63]. The mice used by List et al. were obese and hyperinsulinemic and showed impaired glucose tolerance. These factors may account for the difference in the results. Serum concentration of insulin, especially insulin concentration of portal vein, is an important regulator of hepatic GHR expression in rodents [64,65].

In this study, eight-week-old male CKD or sham mice on c57BL/6J background were given GH or vehicle for 6 weeks, and all mice were sacrificed at the age of 14 weeks old. We showed that GH administration elicited beneficial metabolic effects in CKD mice. c57BL/6J mice are the most widely used inbred strain for biomedical research. For c57BL/6J mice, many developmental processes such as T-cell and B-cell immunity, as well as the central nervous system, are still ongoing until 26 weeks of life [66,67,68]. Furthermore, growth patterns and body composition were evaluated in c57BL/6J mice. Data suggested that cortical bone property and peak bone mass on male c57BL/6J mice are not reached until around 26 weeks of age [69,70,71,72]. Thus, the results of our present study are of immerse importance, as multiple disturbances in the GH/IGF-I axis have been observed in children with CKD. 

We recognize the limitations of this study. Firstly, according to our restrictive study design, vehicle-treated CKD mice were fed ad libitum, whereas other mouse groups received an energy intake amount equal to that of vehicle-treated CKD mice. However, we observed that pair-fed mice consumed their restricted amount of the rodent diet within a short period of time. These pair-fed mice were in an overnight fasting state. Mice, as nocturnal creatures, are active mainly during the dark phase. Circadian rhythm affects adipose tissue metabolism [73,74,75]. Disruption of circadian regulation has been implicated in cancer-induced WAT wasting [42]. Secondly, our work was performed in male c57BJ/6J mice. Results generated from male mice cannot be unambiguously extrapolated to female mice. Sex hormones influence regional adipose tissue fatty acid storage and BAT function in animals and humans. Disruption of estrogen signaling such as by performing ovariectomy resulted in reduced energy expenditure, gain of fat mass, and loss of BAT activity, and these metabolic phenotypes can be reversed by subsequent estrogen replacement in ovariectomized rodents [76]. The reduction of circulating concentration of estradiol is associated with central obesity and decreased metabolism in menopause [77]. Murine and human brown adipocytes express estrogen receptor α [78,79]. Intracerebral administration of estrogen increased BAT activity in mice [80]. On the other hand, follicle-stimulating hormones, which are elevated with estrogen deficiency, downregulated in vivo BAT function in mice [81]. Currently, there are no published data on the effect of estrogen or estrogen deficiency on in vivo BAT function in humans. Dieudonne et al. investigated the effects of sex hormones on adipogenesis in preadipocytes from male versus female rats. They found that androgens and estrogens did not affect adipogenesis in cultured preadipocytes from male rats. However, opposite effects of androgens and estrogens on adipogenesis have been demonstrated in cultured preadipocytes from female rats. Estrogens increased adipogenesis, while androgens acted as negative effectors of terminal differentiation on rat preadipocytes. Subsequent studies suggest that these opposite effects could be related to differential expression of IGF-IR and Pparγ2 on those cultured preadipocytes [82]. Thirdly, uncertainty remains about the precise role of BAT metabolic responses in the pathogenesis of cachexia, and this is partly due to the lack of BAT-specific pharmacological agents. Currently, there is no convincing evidence to suggest that BAT activity can be selectively modulated by any pharmacological agents without influencing WAT metabolism along with cardiac chronotropic side-effects [83,84]. Moreover, BAT activity is mostly driven by the sympathetic signal mediated by β-adrenergic receptors, namely, ADRB3 in mice and ADRB1/ADRB2 in humans. However, the in vivo BAT metabolic activity is the result of the interaction between sympathetic output signal to BAT and other concomitant signaling processes such as α-adrenergic receptors and adenosine receptors as well as postsignaling modulation of these signaling processes [84]. The complexity and redundancy of the endogenous sympathetic regulation of BAT metabolic activity may explain the lack of an optimal pharmacological approach to modulate BAT in vivo. 

Previously, we performed RNAseq analysis in the gastrocnemius muscle in CKD and control mice and identified the top 12 differentially expressed genes that have been associated with energy metabolism, skeletal and muscular system development and function, nervous system development and function, as well as organismal injury and abnormalities [23]. We evaluated the effects of GH treatment on muscle transcriptome in this study. A total of 7 of the 12 muscle genes with variable expression in CKD mice were normalized or reduced by GH (Figure 6). These seven muscle genes—Atp2a2, Cyfip2, Fhl1, Tnnc1, Atf3, Fos, and Itpr1—had aberrant expression patterns that have been linked to enhanced tissue thermogenesis, compromised mechanical muscle properties, poor muscle regeneration, and diminished muscle-neuron regeneration capacity [23]. 

In conclusion, our findings imply that GH might be a useful treatment for adipose tissue browning and muscular atrophy in CKD-associated cachexia.

## 4. Materials and Methods

### 4.1. Study Design

This study was conducted in compliance with established guidelines and the prevailing protocol (S01754) as approved by the Institutional Animal Care and Use Committee (IACUC) at the University of California, San Diego, in accordance with the National Institutes of Health. Recombinant human GH was kindly provided by Genetech (South San Francisco, CA, USA). Six-week-old male c57BL/6J mice were purchased from the Jackson Laboratory (strain: 000664) (Bar Harbor, ME, USA) and used for this study. CKD in mice were induced by 2-stage 5/6 nephrectomy while a sham operation was carried out in control mice [13]. Individual mice were housed in each cage in 12:12 hour light–dark cycles with ad libitum access to mouse diet 5015 (LabDiet, St. Louis, MO, USA, catalog 0001328, with a metabolizable energy value of 3.59 kcal/g) and water prior to the initiation of the experiment. We performed the following two studies. Study 1: We evaluated the dietary effects of GH in CKD and sham mice. CKD and sham mice were administrated with GH (5 mg/kg/day or 10 mg/kg/day, intraperitoneal) or vehicle (normal saline), respectively. The study period was 42 days, and all mice were fed ad libitum. We measured caloric intake and accompanying weight change in CKD and sham mice. The caloric intake for each mouse was calculated by multiplying total mouse diet consumption during the 42 days (in grams) with the metabolizable energy value of the diet (3.59 kcal/g). Average daily energy intake in mice was expressed as kcal/mouse/day. Study 2: We evaluated the effects of GH in CKD mice beyond nutritional stimulation by employing a diet-restrictive strategy. CKD and sham mice were given GH (10 mg/kg/day, intraperitoneal) or vehicle for 42 days. Each mouse was individually housed during the study period. CKD mice treated with vehicle were fed ad libitum. We measured caloric intake in vehicle-treated CKD mice by multiplying total mouse 5015 diet consumption during the 42 days (in grams) with the metabolizable energy value of the diet (3.59 kcal/g). The average daily energy intake for vehicle-treated CKD mice was calculated and expressed as kcal/mouse/day. We then fed the same amount of mouse 5015 diet based on the recorded average daily energy intake for vehicle-treated CKD mice to other groups of mice, i.e., CKD mice treated with GH (10 mg/kg/day, intraperitoneal) as well as sham mice treated with GH (10 mg/kg/day, intraperitoneal) or vehicle. We fed the mice daily during the daytime (0900-1200). We measured weekly weight change for each mouse. The schematic study plan for the ad libitum and diet-restrictive study is illustrated in Figure 1A,D, respectively.

### 4.2. Body Composition, Metabolic Rate, and In Vivo Muscle Function

Body composition (for lean and fat content) was measured by quantitative magnetic resonance analysis (EchoMRI-100^TM^, Echo Medical System, Houston, TX, USA) [13,23]. Resting metabolic rate was assessed by using Oxymax calorimetry (Columbus Instruments, Columbus, OH, USA) during the daytime (0900-1700) [13]. At the end of the study, rotarod activity (model RRF/SP, Accuscan Instrument, Columbus, OH, USA) and forelimb grip strength (Model 47106, UGO Basile, Gemonio, Italy) in mice were assessed [13,23]. 

### 4.3. Serum and Blood Chemistry

Mice were sacrificed and serum samples were collected within 4 h after the last rhGH or vehicle injection. VetScan^®^ Comprehensive Diagnostic Profile reagent rotor and the VetScan Chemistry Analyzer (Union City, CA, USA) were used for quantitative determination of BUN and serum creatine concentration (Appendix A). Concentrations of serum GH in mice were analyzed using commercially available ELISA kits according to the manufacturer’s protocols (Appendix A). 

### 4.4. Protein Assay for Muscle and Adipose Tissue

Portions of the right gastrocnemius muscle, inguinal WAT, and interscapular BAT were processed in a tissue homogenizer (Omni International, Kennesaw, GA, USA). Protein concentration of tissue homogenate was assayed using a Pierce BCA Protein Assay Kit (Thermo Scientific, catalog 23227, Waltham, MA, USA). Uncoupling (UCP) protein content in muscle and adipose tissue homogenates were assayed. In addition, adenosine triphosphate (ATP) content of tissue homogenate was assessed by using an ATP Assay Kit which relies on the phosphorylation of glycerol that could be quantified by colorimetric or fluorometric methods (Abcam, catalog ab83355, Cambridge, UK). Protein concentration of phospho-JAK2 and total JAK2, as well as phospho-STAT5 and total STAT5, in muscle homogenates was measured (Appendix A).

### 4.5. Muscle RNAseq Analysis

Previously, we performed RNAseq analysis on gastrocnemius muscle mRNA in 12-month-old CKD mice versus age-appropriate sham control mice [23]. Detailed procedures for mRNA extraction, purification, and subsequent construction of cDNA libraries as well as analysis of gene expression were published. We then performed ingenuity pathway analysis enrichment tests for those differentially expressed muscle genes in CKD mice versus control mice, focusing on pathways related to energy metabolism, skeletal and muscle system development and function, and organismal injury and abnormalities. We identified the top 12 differentially expressed muscle genes in CKD versus control mice. In this study, we performed qPCR analysis for those top 12 differentially expressed gastrocnemius muscle genes in the different experimental groups.

### 4.6. Quantative Real-Time PCR

Portions of the right gastrocnemius muscle of mice, inguinal WAT, and interscapular BAT were processed by using a tissue homogenizer (Omni International, Kennesaw, GA, USA). Total RNA from tissue homogenate was isolated using TriZol (Life Technology, Carlsbad, CA, USA). Total RNA (3 µg) was reverse transcribed to cDNA with SuperScript III Reverse Transcriptase (Invitrogen, Waltham, MA, USA). Quantitative real-time RT-PCR of target genes was performed using KAPA SYBR FAST qPCR kit (KAPA Biosystems, Wilmington, MA, USA) [23]. Glyceraldehyde−3-phosphate dehydrogenase (GAPDH) was used as an internal control. Expression levels were calculated according to the relative 2^−ΔΔCt^ method. All primers are listed (Appendix A).

### 4.7. Statistics

Statistical analyses were performed using GraphPad Prism version 9.4.1 (GraphPad Software, San Diego, CA, USA). All data are presented as mean ± S.E.M. For comparison of the means between two groups, data were analyzed by Student’s 2-tailed *t*-test. Differences of the means for more than two groups containing two variables were analyzed using 2-way ANOVA. Posthoc analysis was performed with Tukey’s test. A *p*-value of less than 0.05 was considered significant.

## Figures and Tables

**Figure 1 ijms-23-15310-f001:**
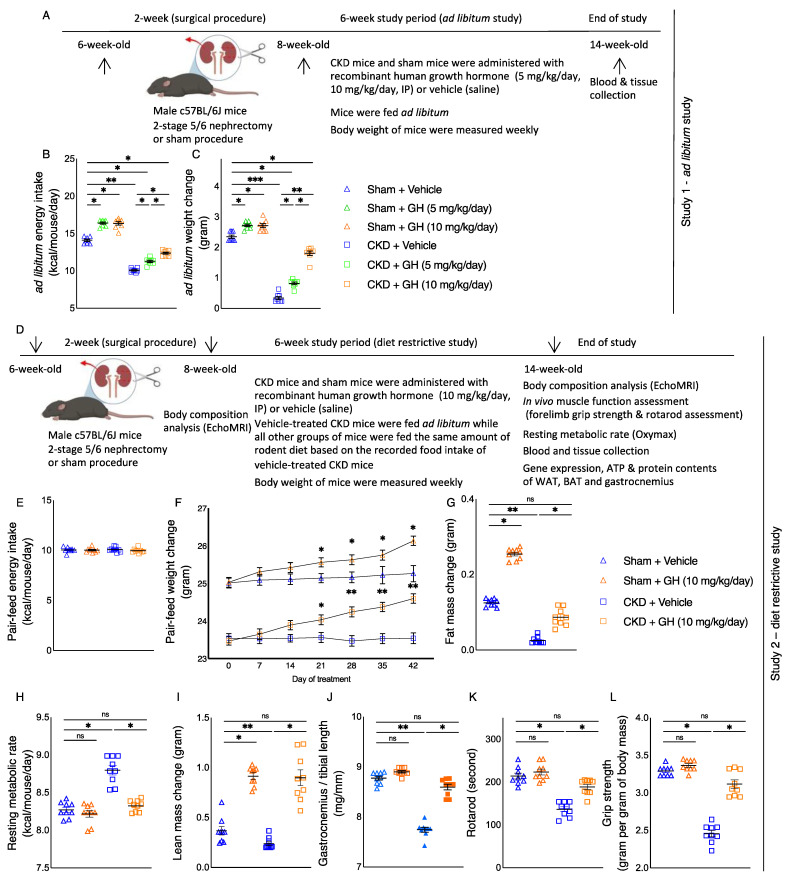
GH attenuates cachexia in CKD mice. We performed two studies. For the first study, we used ad libitum dietary strategy (**A**). CKD and control mice were given GH (5 mg/kg/day or 10 mg/kg/day), or vehicle (normal saline), respectively, for six weeks. All mice were fed ad libitum. We calculated average daily caloric intake (**B**) and recorded final weight change in mice (**C**). Results of Sham + GH (5 mg/kg/day) and Sham + GH (10 mg/kg/day) mice were compared to those of Sham + Vehicle mice, while results of CKD + GH (5 mg/kg/day) and CKD + GH (10 mg/kg/day) mice were compared to those of CKD + Vehicle mice. In addition, results of CKD + GH (5 mg/kg/day) mice were compared to those of CKD + GH (10 mg/kg/day) mice. Furthermore, results of CKD + Vehicle, CKD + GH (5 mg/kg/day), and CKD + GH (10 mg/kg/day) mice were compared to those of Sham + Vehicle, Sham + GH (5 mg/kg/day), and Sham + GH (10 mg/kg/day) mice, respectively. Data are expressed as mean ± SEM. For comparison of the means between two groups, data were analyzed by Student’s 2-tailed *t*-test. Differences of the means for more than two groups containing two variables were analyzed using two-way ANOVA. Posthoc analysis was performed with Tukey’s test. Specific *p*-values are shown above the bar. * *p* < 0.05, ** *p* < 0.01, *** *p* < 0.001. ns signifies not significant. For the second experiment, we employed a diet-restrictive strategy (**D**). CKD + Vehicle mice were given an ad libitum amount of food, whereas other groups of mice were given an equivalent amount of food (**E**). Weight gain, fat content, resting metabolic rate, lean content, gastrocnemius weight relative to length of tibia, and in vivo muscle function (rotarod and grip strength) were measured (**F**–**L**). Results of Sham + GH (10 mg/kg/day) mice were compared to those of Sham + Vehicle mice, while results of CKD + GH (10 mg/kg/day) mice were compared to those of CKD + Vehicle mice. Furthermore, results of CKD + Vehicle and CKD + GH (10 mg/kg/day) mice were compared to those of Sham + Vehicle mice, respectively. Data are expressed as mean ± SEM. For comparison of the means between two groups, data were analyzed by Student’s 2-tailed *t*-test. Posthoc analysis was performed with Tukey’s test. Specific *p*-values are shown above the bar. ns signifies not significant, * *p* < 0.05, ** *p* < 0.01.

**Figure 2 ijms-23-15310-f002:**
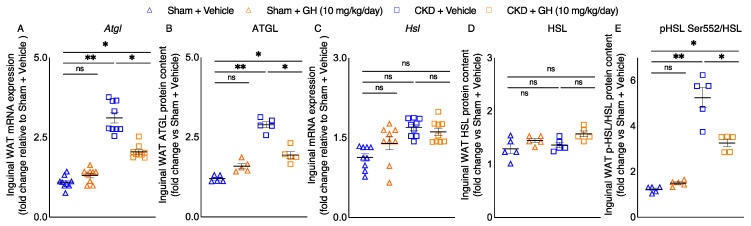
GH enhances energy balance in skeletal muscle and adipose tissue. Measurements were made of the UCP (**A**,**C**,**E**) and ATP contents (**B**,**D**,**F**) in gastrocnemius, WAT, and BAT. CKD mice were fed ad libitum, whereas other mouse groups received an energy intake amount equal to that of CKD + Vehicle mice. Comparisons were made between the outcomes of Sham + GH (10 mg/kg/day) mice and Sham + Vehicle mice, as well as between the outcomes of CKD + GH (10 mg/kg/day) mice and CKD + Vehicle mice. Additionally, the outcomes of the CKD + Vehicle and CKD + GH (10 mg/kg/day) mice were contrasted with those of the Sham + Vehicle mice. Data are expressed as mean ± SEM. For comparison of the means between two groups, data were analyzed by Student’s 2-tailed *t*-test. Posthoc analysis was performed with Tukey’s test. Specific *p*-values are shown above the bar. ns signifies not significant, * *p* < 0.05, ** *p* < 0.01.

**Figure 3 ijms-23-15310-f003:**
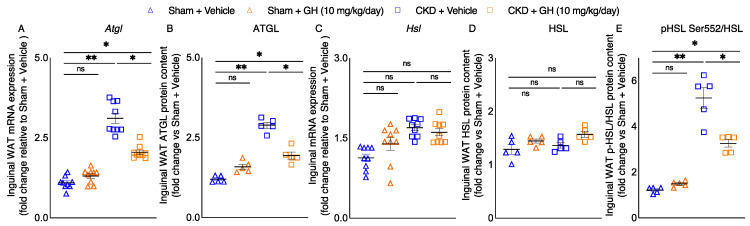
Lipolytic gene expression and protein content in CKD mice. CKD mice were fed ad libitum, whereas other mouse groups received an energy intake amount equal to that of CKD + Vehicle mice. By using qPCR, the expression of lipolytic genes (Atgl and Hsl) in the inguinal WAT was determined (**A**,**C**). In addition, the total protein content of ATGL and HSL as well as the relative phosphorylated HSL/total HSL ratio in the inguinal WAT were evaluated (**B**,**D**,**E**). Results are analyzed and expressed as in Figure 2. Data are expressed as mean ± SEM. For comparison of the means between two groups, data were analyzed by Student’s 2-tailed *t*-test. Posthoc analysis was performed with Tukey’s test. Specific *p*-values are shown above the bar. ns signifies not significant, * *p* < 0.05, ** *p* < 0.01.

**Figure 4 ijms-23-15310-f004:**
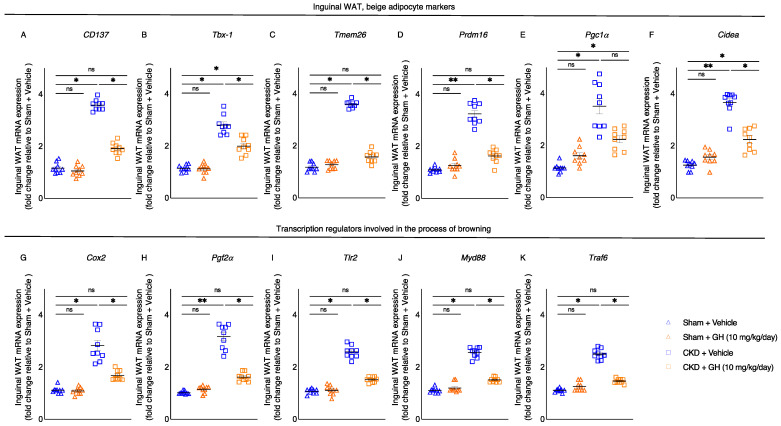
GH reduces browning of adipose tissue in CKD mice. CKD mice were fed ad libitum, whereas other mouse groups received an energy intake amount equal to that of CKD + Vehicle mice. qPCR was used to assess the gene expression of the beige adipocyte markers CD137, Tbx-1, Tmem26, Prdm16, Pgc1α, and Cidea in the inguinal WAT (**A**–**F**). In addition, inguinal WAT was also used to evaluate the gene expression of the Cox2 signaling pathway and the toll-like receptor pathway (Cox2, Pgf2, Tlr2, Myd88, and Traf6) (**G**–**K**). Final results were expressed in arbitrary units, with one unit being the mean level in Sham + Vehicle mice. Results are analyzed and expressed as in Figure 2. Data are expressed as mean ± SEM. For comparison of the means between two groups, data were analyzed by Student’s 2-tailed *t*-test. Posthoc analysis was performed with Tukey’s test. Specific *p*-values are shown above the bar. ns signifies not significant, * *p* < 0.05, ** *p* < 0.01.

**Figure 5 ijms-23-15310-f005:**
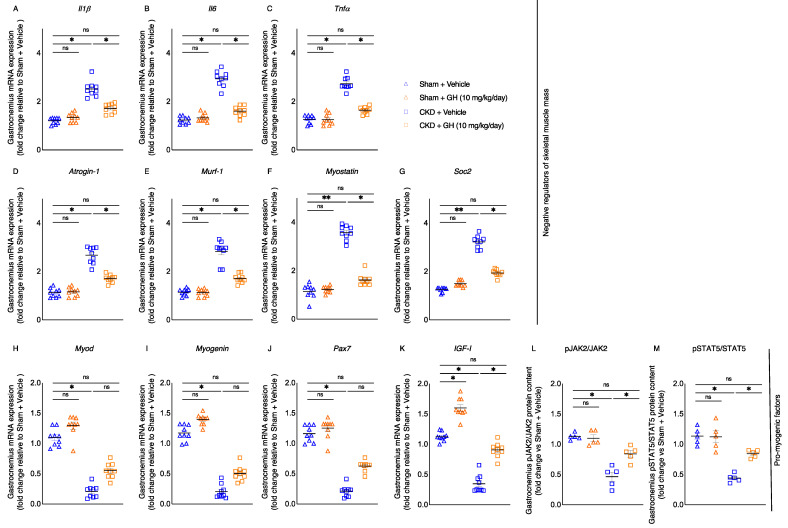
GH reduces muscle-wasting signaling pathways in CKD mice. CKD mice were fed ad libitum, whereas other mouse groups received an energy intake amount equal to that of CKD + Vehicle mice. By using qPCR, the expression of negative regulators of skeletal muscle mass (Il1β, Il6, Tnfα, Atrogin-1, Murf-1, and Socs2) as well as promyogenic factors (MyoD, Myogenin, Pax7, and IGF-I) in the gastrocnemius muscle was determined (**A**–**K**). In addition, by using the appropriate ELISA kits, the relative phosphorylated JAK2/total JAK2 ratio and the phosphorylated STAT5/total STAT5 ratio in the gastrocnemius muscle were evaluated (**L**,**M**). Results are analyzed and expressed as in Figure 2. Data are expressed as mean ± SEM. For comparison of the means between two groups, data were analyzed by Student’s 2-tailed *t*-test. Posthoc analysis was performed with Tukey’s test. Specific *p*-values are shown above the bar. ns signifies not significant, * *p* < 0.05, ** *p* < 0.01.

**Figure 6 ijms-23-15310-f006:**
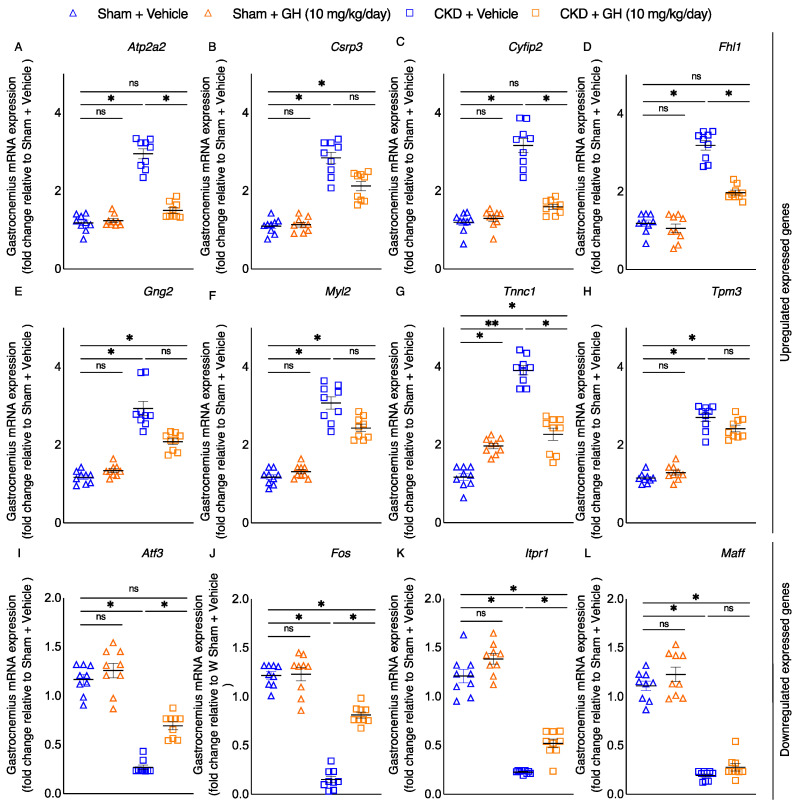
GH reduces the expression of differentially expressed genes in the muscles of CKD mice. CKD mice were fed ad libitum, whereas other mouse groups received an energy intake amount equal to that of CKD + Vehicle mice. The expression of relevant genes in the mouse gastrocnemius muscle was assessed using qPCR (**A**–**L**). Results are analyzed and expressed as in Figure 2. Data are expressed as mean ± SEM. For comparison of the means between two groups, data were analyzed by Student’s 2-tailed *t*-test. Posthoc analysis was performed with Tukey’s test. Specific *p*-values are shown above the bar. ns signifies not significant, * *p* < 0.05, ** *p* < 0.01.

**Table 1 ijms-23-15310-t001:** Serum and blood chemistry of mice from ad libitum study. Eight-week-old CKD and sham mice were treated with GH (5 mg/kg per day or 10 mg/kg per day) or normal saline as a vehicle for six weeks. All mice were fed ad libitum. Data are expressed as mean ± SEM. Results of all five groups of mice were compared to those of Sham + Vehicle mice, respectively. BUN, blood urea nitrogen. ^a^ *p* < 0.05, significantly higher than Sham + Vehicle mice.

	Sham + Vehicle (*n* = 9)	Sham + GH (5 mg/kg/day) (*n* = 9)	Sham + GH (10 mg/kg/day) (*n* = 9)	CKD + Vehicle (*n* = 9)	CKD + GH (5 mg/kg/day) (*n* = 9)	CKD + GH (10 mg/kg/day) (*n* = 9)
BUN (mg/dL)	34.5 ± 3.5	36.7 ± 4.6	32.6 ± 3.7	65.8 ± 6.9 ^a^	75.6 ± 8.1 ^a^	65.9 ± 5.8 ^a^
Creatinine (mg/dL)	0.32 ± 0.11	0.35 ± 0.14	0.28 ± 0.09	0.57 ± 0.15 ^a^	0.65 ± 0.13 ^a^	0.75 ± 0.13 ^a^

**Table 2 ijms-23-15310-t002:** Serum and blood chemistry in mice from diet-restrictive study. Eight-week-old CKD and sham mice were treated with GH (10 mg/kg per day), or normal saline as a vehicle for six weeks. CKD mice were fed ad libitum. The other mouse groups received an energy intake amount equal to that of CKD + Vehicle mice. BUN, blood urea nitrogen. Results are analyzed and presented as in Table 1. ^a^ *p* < 0.05, significantly higher than Sham + Vehicle mice.

	Sham + Vehicle (*n* = 9)	Sham + GH (10 mg/kg/day) (*n* = 9)	CKD + Vehicle (*n* = 9)	CKD + GH (10 mg/kg/day) (*n* = 9)
BUN (mg/dL)	36.5 ± 5.8	26.7 ± 4.7	59.8 ± 7.4 ^a^	72.8 ± 11.5 ^a^
Creatinine (mg/dL)	0.25 ± 0.06	0.31 ± 0.13	0.63 ± 0.21 ^a^	0.75 ± 0.25 ^a^
Human GH (µg/L)	-	364.6 ± 76.4	-	325.3 ± 65.3

## Data Availability

The authors confirm that the data supporting the findings of this study are available within the article and its Appendix A. Additional raw data supporting the findings of this study are available from the corresponding author (R.H.M.) on request.

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
