# Peer review of "Growth Hormone Improves Adipose Tissue Browning and Muscle Wasting in Mice with Chronic Kidney Disease-Associated Cachexia"

_ijms, 2022, doi:10.3390/ijms232315310_

Round 1
Reviewer 1 Report
In this paper “Growth hormone improves adipose tissue browning and muscle wasting in chronic kidney disease-associated cachexia” Mak et al. show that in a mouse model of chronic kidney disease, there is a metabolic shift in the animals with weight reductions that can be counteracted by growth hormone. They provide some gene and protein expression data from muscle and adipose tissue to support the claim that GH protects from CKD-associated cachexia.
It seems like a solid study, using one mouse model to prove a point. The introduction is short and concise, in my view somewhat unbalanced towards own publications. The results are written in a very understandable manner, but more explanations would help the reader, for instance, rather than writing “the results of the blood biochemistry are shown in Tab1” it would be helpful to describe the result of what this Table shows. The methods require a major improvement, as in the current form, it would be impossible to repeat the experiments outlined here.
Therefore, overall the data are relevant, but the manuscript needs some major improvements before considering publication.
1) General concerns
- Since this is exclusively a study on 1 mouse model of CKD, it is mandatory to put the word „mouse/mice“ in the title to avoid a misleading title
- The functional relevance of brown and beige adipose tissue in humans in general, but in particular in the cachexia context, is very questionable. Hence the conclusion of the paper (GH counteracts browning and therefore improves cachexia) may perhaps be relevant for mice but certainly not for humans. Even in mice, the appearance and functional relevance of UCP1-related thermogenesis in cachexia is highly debated (see for instance PMID: 27571348, PMID: 35166050). This needs to be clearly stated.
- GH resistance seems to play a particularly big role in children with CKD. To what extent is it relevant that very young mice (6 weeks) were studied here? It should be noted as limitation of the study that it is unclear how adult mice would deal with GH in this setting.
2) Results
- Is any muscle atrophy present? A relevant quantification of this should be added: Size/weight of muscle, fibre size? It is surprising not to see lean mass changes despite the increase in atrogene expression
- Why is there no change in lean mass, opposite to the authors' previous publication using the identical mouse model? (PMID: 32843714)
- Specify the abbreviation BUN in Tab1/2
- CD137, Tbx1, and Tmem26 are not classical beige fat markers. These are Prdm16, Pgc1a, Cidea, Ucp1, which should be shown if “browning” is claimed
- Lipolysis is described to be increased in cachexia, hence a characterization of this pathway (e.g. circulating lipids/free fatty acids, expression of key lipases in adipose tissue) would be relevant
- “Results are analyzed and expressed in Figure 2” makes no sense. Should it be “as in Figure 2”? Also, please specify if pair-fed or random fed, and what statistical test was used, for each figure.
- Given the crosstalk of the GH and IGF pathways, it would be interesting to learn more about the interaction in the CKD setting. Is the GH effect dependent on the elevated IGF1 expression? Are circulating IGF levels altered?
3) Methods
In general, the methods need to be written in a much more detailed manner, as they cannot be reproduced with the descriptions given here. Some examples are given below, but a general improvement is necessary!
- Specify if Bl6J or N mice were used to study; exact order number needs to be given.
- When did treatment with GH start, at which time after surgery?
- More info on pair-feeding is necessary. Were mice matched 1:1 to a CKD mouse, or did authors use average of all CKD. When was diet given, once daily or smaller doses throughout the day? At which time? Mice with restricted access to food will eat everything at once, so they effectively fast 23hours. How does this relate to CKD? How is circardian rhythm affected by this treatment?
- Blood chemistry: Specify “standard methods”
- Typo in Pierce BCA assay kit
- Protein: Specify antibodies and methods used for ATP measurement.
4) Discussion
- “We also demonstrated that the beneficial metabolic benefits of GH go beyond appetite stimulation.” This conclusion is only correct when looking at the data of Fig1, all molecular analyses were performed in the non-pairfed mice, so might be secondary to improved energy intake
- “Finally, using RNAseq analysis, we assessed the muscle transcriptome” this statement is not correct as it was done in a previous study
- “thermogenesis” in muscle, assuming this based on UCP3 expression is incorrect. UCP3 is expressed in low levels, and does not have the same role as UCP1 (for instance, see PMID: 31133866). So this is incorrect wording, in addition to unclear functional relevance. Many functions of UCP3 are known which could contribute to muscle dysfunction in cachexia, and could be discussed instead
Author Response
We appreciate the insightful comments from reviewers and we have revised the manuscript according to their constructive comments. Below is a point-by-point reply to their specific comments. Our additional information in this revised version has been highlighted in blue color.
Comments from reviewer 1
- Since this is exclusively a study on 1 mouse model of CKD, it is mandatory to put the word „mouse/mice“ in the title to avoid a misleading title]
We agree with the comment and new title is ‘Growth hormone improves adipose tissue browning and muscle wasting in mice with chronic kidney disease-associated cachexia’
- The functional relevance of brown and beige adipose tissue in humans in general, but in particular in the cachexia context, is very questionable. Hence the conclusion of the paper (GH counteracts browning and therefore improves cachexia) may perhaps be relevant for mice but certainly not for humans. Even in mice, the appearance and functional relevance of UCP1-related thermogenesis in cachexia is highly debated (see for instance PMID: 27571348, PMID: 35166050). This needs to be clearly stated.
We agree with the comments. Additional information has been provided in the section of discussion (2nd paragraph of discussion, page 9). Specifically, the important issues raised by both published data from PMID: 27571348 and PMID: 35166050 have been incorporated as our reference #31 and #32, respectively, for this revised manuscript.
Detailed information is the following, ‘By increasing muscle mass and improving energy efficiency, GH may improve muscle strength. Anaerobic and aerobic energy sources comprise the continuum of energy needed to fuel muscular function. Anaerobic energy systems are stimulated by GH, which suppresses the aerobic energy system. This increases muscle strength. After six months of GH therapy, healthy males showed a considerable improvement in their lower body muscle strength [5]. By "uncoupling" ATP synthesis, UCPs regulate energy homeostasis by dissipating the mitochondrial proton gradient for ATP synthesis and producing heat [25,26]. UCP3 is expressed in skeletal muscle and upregulation of UCP3 has been reported in various conditions characterized by skeletal muscle atrophy, including denervation, diabetes, cancer and sepsis [27]. Gastrocnemius UCP3 protein content along with ATP content was normalized in GH-treated CKD mice (Figure 4, A & B). Putative functions of UCP3 are controversial. Interesting evidence for and against UCP3 involvement in thermogenesis has been published [27,28]. Furthermore, increased muscle expression of UCP3 has been postulated to modulate oxidative stress and lipotoxicity in rat model of cachexic sepsis [29]. GH therapy reduced abnormal UCP1 and ATP content in WAT and BAT in CKD mice (Figure 4, C-F). However, the precise role of UCP1 in disease-associated cachexia in human is still a topic of debate. Several studies have described UCP1 expression, a biomarker of WAT browning, as a critical component of WAT dysfunction in cancer cachexia [17,18,30]. Results also suggested that an UCP-1 independent cascade could also regulate adipocyte homeostasis and influence tumor-induced WAT wasting [31]. Moreover, activation of BAT has been associated with hypermetabolism in cachexia but information from human study is scarce. A recent study investigated the relationship between activation of BAT and hypermetabolism in patients with emphysematous COPD (chronic obstructive pulmonary disease). BAT activity and gene expression of beige markers of BAT in WAT (Tmem26, Cidea, CD137, Shox2 and Tnfrsf9) were not different between COPD patients versus controls [32]. Medications may influence the sympathetic nerve system and BAT metabolism. Adrenergic receptor blockers and calcium channel blockers are commonly used by COPD patients. Involvement of b-adrenergic receptor signaling in BAT metabolism was reported in humans and rodents [33,34]. Data also indicated that calcium channel blockers regulated adipogenesis and BAT browning [35,36].
- GH resistance seems to play a particularly big role in children with CKD. To what extent is it relevant that very young mice (6 weeks) were studied here? It should be noted as limitation of the study that it is unclear how adult mice would deal with GH in this setting.
We have addressed this issue. Information has been provided on page 12, ‘In this study, eight-week-old male CKD or sham mice on c57BL/6J background were given GH or vehicle for 6 weeks and all mice were sacrificed at the age of 14-week-old. We showed that GH administration elicited beneficial metabolic effects in CKD mice. C57BL/6J mice are the most widely used inbred stain for biomedical research. For c57BL/6J mice, many developmental processes such as T-cell and B-cell immunity as well as central nervous system are still ongoing until 26 weeks of life [67-69]. Furthermore, growth patterns and body composition were evaluated in c57BL/6J mice. Data suggested that cortical bone property and peak bone mass on male c57BL/6J are not reached until around 26 weeks of age [70-73]. Thus, results of our present study are of immerse importance as multiple disturbances in the GH/IGF-I axis has been observed in children with CKD.’
We also provide the relevant information for this revised manuscript, on page 12-13
The GH dose administered to the mice in this study is about 200-fold higher than the dose typically used in humans. The recommended dose approved for treatment of growth failure in children with CKD is 0.35 mg/kg per week [59] whereas we used 10 mg/kg/day in mice for this study. However, our dose was comparable to those commonly used in rodent studies [60], and the observation of increased muscle mRNA expression of IGF-I as well as JAK/STAT phosphorylation after GH treatment in our CKD mice (Figure 7, L-M) argues against any effect of GHR saturation.
We recognize the limitations of this study. Firstly, according to our restrictive study design, vehicle-treated CKD mice were fed ad libitum whereas other mouse groups received an energy intake amount equal to that of vehicle-treated CKD mice. However, we observed that pair-fed mice consumed their restricted amount of rodent diet within a short period of time. These pair-fed mice were in an overnight fasting stage. Mouse, as nocturnal creatures, are active mainly during the dark phase. Circadian rhythm affects adipose tissue metabolism [74-76]. Disruption of circadian regulation has been implicated in cancer-induced WAT wasting [42]. Secondly, our work was performed in male c57BJ/6J mice. Results generated from male mice cannot be unambiguously extrapolated to female mice. Sex hormones influence regional adipose tissue fatty acid storage and BAT function in animals and humans. Disruption of estrogen signaling such as by performing ovariectomy resulted in reduced energy expenditure, gain of fat mass and loss of BAT activity and these metabolic phenotypes can be reversed by subsequent estrogen replacement in ovariectomized rodents [77]. The reduction of circulating concentration of estradiol is associated with central obesity and decreased metabolism in menopause [78]. Murine and human brown adipocytes express estrogen receptor a [79,80]. Intracerebral administration of estrogen increased BAT activity in mice [81]. On the other hand, follicle-stimulating hormone, which is elevated with estrogen deficiency, downregulated in vivo BAT function in mice [82]. Currently, there is no published data on the effect of estrogen or estrogen deficiency on in vivo BAT function in humans. Dieudonne MN, et al investigated the effects of sex hormones on adipogenesis in preadipocytes from male versus female rats. They found that androgens and estrogens did not affect adipogenesis in cultured preadipocytes from male rats. However, opposite effects of androgens and estrogens on adipogenesis has been demonstrated in cultured preadipocytes from female rats. Estrogens increased adipogenesis while androgens acted as negative effector of terminal differentiation on rat preadipocytes. Subsequent studies suggest that these opposite effects could be related to differential expression of IGF-IR and Pparg2 on those cultured preadipocytes [83]. Thirdly, uncertainty remains about the precise role of BAT metabolic responses in the pathogenesis of cachexia and this is partly due to the lack of BAT-specific pharmacological agents. Currently, there is no convincing evidence to suggest that BAT activity can be selectively modulated by any pharmacological agents without influencing WAT metabolism along with cardiac chronotropic side-effects [84,85]. Moreover, BAT activity is mostly driven by the sympathetic signal mediated by b-adrenergic receptors; ADRB3 in mice and ADRB1/ADRB2 in humans. However, the in vivo BAT metabolic activity is the result of the interaction between sympathetic output signal to BAT and other concomitant signaling processes such as a-adrenergic receptors and adenosine receptors as well as post-signaling modulation of these signaling processes [85]. The complexity and redundancy of the endogenous sympathetic regulation of BAT metabolic activity may explain the lack of optimal pharmacological approach to modulate BAT in vivo.
- Is any muscle atrophy present? A relevant quantification of this should be added: Size/weight of muscle, fibre size? It is surprising not to see lean mass changes despite the increase in atrogene expression. Why is there no change in lean mass, opposite to the authors' previous publication using the identical mouse model? (PMID: 32843714)
We have showed a significant decreased of gastrocnemius weight in Figure 2H. Change in lean mass is shown in Figure 2G.
- Specify the abbreviation BUN in Tab1/2
We have listed the abbreviation for BUN in Tables 1 & 2.
- CD137, Tbx1, and Tmem26 are not classical beige fat markers. These are Prdm16, Pgc1a, Cidea, Ucp1, which should be shown if “browning” is claimed.
Information for Prdm16, Pgc1a, Cidea, Ucp1 gene expression is shown in Figure 6, D-F.
- Lipolysis is described to be increased in cachexia, hence a characterization of this pathway (e.g. circulating lipids/free fatty acids, expression of key lipases in adipose tissue) would be relevant.
Data on lipolytic enzymes in CKD mice were provided in section 2.4.
2.4. GH mitigates lipolytic enzymes in CKD mice
Elevated lipolysis is important for adipose tissue wasting in cachexia [21]. We investigated the molecular basis for the loss of adipose tissue in CKD mice. Inguinal WAT gene expression and protein content of adipose triglyceride lipase (ATGL) was significantly increased in vehicle-treated CKD mice (Figure 5, A&B). Inguinal WAT gene expression and protein content of hormone sensitive lipase (HSL) was not different among groups of mice (Figure 5, C&D). However, phosphorylated HSL Ser552 protein content in inguinal WAT, a surrogate marker for protein kinase A activated lipolysis, was 5-fold higher in vehicle-treated CKD mice compared to control mice (Figure 5E). Importantly, GH significantly decreased inguinal WAT gene expression and protein content of ATGL as well as protein content of phosphorylated HSL in CKD mice.
Relevant information also provides in the section of discussion, page 10/11, ‘CKD-associated cachexia is a progressive, multi-factorial metabolic syndrome that results in significant loss of adipose tissue and skeletal muscle mass. Fat loss from adipose tissue in CKD-associated cachexia may be due to the increased rate of lipolysis. Recent longitudinal studies found that magnitude of adipose tissue wasting predicts poorer survival in cancer patients [37-39]. The bulk of lipid mobilization from adipose tissue is mediated through lipolysis. In canonical adipose tissue lipolysis, triglycerides stored in lipid droplets are hydrolyzed by ATGL and HSL to produce free glycerol and fatty acids and fuel peripheral tissue metabolism [40]. ATGL is the rate limiting lipase that is responsible to hydrolyze triacylglycerol in lipid droplets to diacylglycerol. GH treatment attenuated inguinal WAT mRNA expression and protein content of ATGL in CKD mice (Figure 5, A&B). Previous studies have shown increased ATGL expression in the adipose tissue of cancer-associated cachectic animals and human [21,41,42]. Inguinal WAT gene expression and protein content of HSL was not different among groups of mice (Figure 5, C&D). However, phosphorylated HSL Ser552 protein content in inguinal WAT, a surrogate marker for protein kinase A activated lipolysis [43], was significantly increased in CKD mice (Figure 5E). Importantly, GH attenuated inguinal WAT protein content of phosphorylated HSL in CKD mice. Evidence of enhanced protein kinase A – activated lipolysis correlated with elevated whole-organism energy expenditure and increased adipose tissue thermogenesis as well as increased expression of biomarkers of adipose tissue browning in WAT was reported in a mouse model of cancer cachexia [21]. Moreover, increased WAT protein content of phosphorylated HSL and protein Kinase A was also shown in mouse model of CKD [44]’.
- Results are analyzed and expressed in Figure 2” makes no sense. Should it be “as in Figure 2”? Also, please specify if pair-fed or random fed, and what statistical test was used, for each figure.
Changes are marked in blue in the revised manuscript.
- Given the crosstalk of the GH and IGF pathways, it would be interesting to learn more about the interaction in the CKD setting. Is the GH effect dependent on the elevated IGF1 expression? Are circulating IGF levels altered?
We have measure serum concentration of recombinant GH (table 2)
|
Sham + Vehicle (n = 9)
|
Sham + GH (10 mg/kg/day) (n = 9)
|
CKD + Vehicle (n = 9)
|
CKD + GH (10 mg/kg/day) (n = 9)
|
BUN (mg/dL) |
36.5 ± 5.8 |
26.7 ± 4.7 |
59.8 ± 7.4 a |
72.8 ± 11.5 a |
Creatinine (mg/dL) |
0.25 ± 0.06 |
0.31 ± 0.13 |
0.63 ± 0.21 a |
0.75 ± 0.25 a |
Human GH (µg/L) |
- |
364.6 ± 76.4 |
- |
325.3 ± 65.3 |
We have explained the reason as why we do not measure serum IGF-I levels (page 12). ‘IGF-I is the most important downstream mediator of GH. Thus, IGF-I is generally considered to be the most important biomarker of GH action as reflected in the inclusion of serum IGF-I in the current guidelines for diagnosis and treatment of GH disorders in humans [61]. We have not measured serum IGF-I concentration in this study as recent studies suggested that concentration of serum IGF-I is not a reliable marker for exogenous growth hormone activity in mice [62]. Male and female mice from four different strains of mice, including 57BL/6J mouse strain used in this study, were treated with recombinant human GH (500 ug/day, intraperitoneally, for a period of 14 days). The total amount of GH administrated to mouse is ~ 7 mg in their study, which is comparable to the dose we used for diet-restrictive study (total amount of GH is ~ 9.2 mg, presumably 22 gram of body weight for CKD mice). In agreement to our observation (serum concentration of human GH in Table 2), GH treatment resulted in high circulating levels of human GH in all four strains of mice whereas no human GH was detectable in control mice receiving isotonic 0.9% NaCl as vehicle. Two-weeks of daily GH treatment significantly increased body and organ weight in male and female mice of all four inbred mouse strains when compared with controls. GH treatment failed to affect circulating (total) IGF-I concentrations in all strains and in both sexes. The liver is the main source of circulating IGF-I [63]. Hepatic expression of IGF-I mRNA did not show any difference between GH-treated mice vs control mice in any of these four strains of mice and sexes [62]. List et alinvestigated the effects of GH in a mouse model of diet-induced diabetes [64]. Male c57BL/6J mice were fed a high-fat diet to induce obesity and type 2 diabetes. Subsequently, obese and diabetic mice were treated with various dose of GH for a period of six-weeks. Comparable to our findings in CKD mice (Figure 2G), their highest dose of GH (215 mg/day/mouse versus 220 mg/day/mouse in our study, presumable 22 gram of body weight for CKD mice) resulted in significantly increment of lean mass content. However, in contrast to the findings that treatment of GH did not influence serum concentration of IGF-I in mice [62], the highest dose of GH used in the study led to a significant increase in serum IGF-I in diabetic mice [64]. The mice used by List et al were obese and hyperinsulinemic and showed impaired glucose tolerance. These factors may account for the difference in the results. Serum concentration of insulin, especially insulin concentration of portal vein, is an important regulator of hepatic GHR expression in rodent [65,66].’
- Methods - In general, the methods need to be written in a much more detailed manner, as they cannot be reproduced with the descriptions given here. Some examples are given below, but a general improvement is necessary. Specify if Bl6J or N mice were used to study; exact order number needs to be given. When did treatment with GH start, at which time after surgery? More info on pair-feeding is necessary. Were mice matched 1:1 to a CKD mouse, or did authors use average of all CKD. When was diet given, once daily or smaller doses throughout the day? At which time? Mice with restricted access to food will eat everything at once, so they effectively fast 23hours. How does this relate to CKD? How is circardian rhythm affected by this treatment?
We have carefully revised the manuscript and additional information is heighted in blue color. New information is listed as shown in the followings;
2.1. GH stimulates food intake and increases body weight in CKD mice
We empirically determined the optimal dose of GH treatments in our mouse model of CKD. Six-week-old c57BL/6J male mice were used for this study. Schematic representation of the experimental design is shown in Figure 1.
Figure 1: Schematic representation for ad libitum study design.
CKD in mice was induced by a two-stage sub-total nephrectomy while sham procedure was performed in control mice [13]. GH treatment was initiated in eight-week-old CKD or sham mice. CKD or sham mice were treated with recombinant human GH (5 mg/kg/day or 10 mg/kg/day, intraperitoneal), or vehicle for six weeks. During the treatment, all mice were housed in individual cage and fed ad libitum. Dietary intake as well as weight gain for each mouse was recorded weekly. Mice were sacrificed at the age of 14-week-old. Serum and blood chemistry of CKD and sham mice are listed (Table 1). CKD mice were uremic as CKD mice had higher concentration of BUN and serum creatinine than control mice. Over the course of six-week ad libitum experiment, GH stimulated food intake and improved weight gain in both CKD and sham mice. GH-treated CKD and GH-treated sham mice exhibited significant average daily energy intake and weight gain than vehicle-treated CKD and vehicle-treated sham mice, respectively (Figure 2, A-B). More importantly, we found that CKD mice treated with 10 mg/kg/day demonstrated significantly improved food intake and weight gain relative to CKD mice treated with 5 mg/kg/day or vehicle. As a result, daily dosing of 10 mg/kg of GH for CKD mice was selected for the subsequent food restrictive study.
2.2. GH improves energy homeostasis in CKD mice
In a second series of experiments, we utilized a food restrictive strategy to study the pharmacological effects of GH in CKD mice beyond appetite stimulation and their consequent body weight gain (Figure 3).
Figure 3: Schematic representation for diet restrictive study design.
Two-stage sub-total nephrectomy for CKD mice and sham procedure for control mice was also performed. Eight-week-old CKD or sham mice were housed individually. Mice were given GH (10 mg/kg/day, intraperitoneal), or vehicle for six weeks. For this diet restrictive study, only vehicle-treated CKD mice were fed ad libitum while the other mouse groups (GH-treated CKD mice as well as GH-treated or vehicle-treated sham mice) received an energy intake amount equal to that of vehicle-treated CKD mice (Figure 2C). All mice were sacrificed at the age of 14-week-old. Serum and blood chemistry are listed in Table 2. Vehicle or GH-treated CKD mice were uremic as they had a higher concentration of BUN and serum creatinine than sham mice. We verified that daily GH treatments resulted in high circulating levels of human GH in mice. Mean circulating human GH was not different between GH-treated CKD (325.3 ± 65.3 µg/L) and GH-treated control mice (364.6 ± 76.4 µg/L) whereas no human GH was detected in CKD or control mice receiving vehicle. Significant increase of weight gain in GH-treated CKD mice relative to vehicle-treated CKD mice was observed at day 21 and the trend remained significant for the rest of the study (Figure 2D). In addition, GH normalized fat and lean mass content, weight of gastrocnemius, resting metabolic rate, and in vivo muscle function (rotarod activity and grip strength) in CKD mice (Figure 2, E–J).
4. Materials and Methods
4.1. Study design
This study was conducted in compliance with established guidelines and prevailing protocol (S01754) as approved by the Institutional Animal Care and Use Committee (IACUC) at the University of California, San Diego in accordance with the National Institutes of Health. Recombinant human GH is kindly provided by Genetech. Six-week-old male c57BL/6J mice were purchased from the Jackson Laboratory (strain: 000664) and used for this study. CKD in mice were induced by 2-stage 5/6 nephrectomy while sham operation was carried out in control mice [13]. Individual mouse was housed in each cage in 12:12 hour light-dark cycles with ad libitum access to mouse diet 5015 (LabDiet, St Louis, MO, USA, catalog 0001328, with a metabolizable energy value of 3.59 kcal/g) and water prior to the initiation of the experiment. We have performed the following two studies. Study 1—we evaluated the dietary effects of GH in CKD and sham mice. CKD and sham mice were administrated with GH (5 mg/kg/day or 10 mg/kg/day, intraperitoneal) or vehicle (normal saline), respectively. The study period was 42 days and all mice were fed ad libitum. We measured caloric intake and accompanying weight change in CKD and sham mice. The caloric intake for each mouse was calculated by multiplying total mouse diet consumption during the 42 days (in grams) with the metabolizable energy value of the diet (3.59 kcal/gram). Average daily energy intake in mice was expressed as kcal/mouse/day. Study 2—we evaluated the effects of GH in CKD mice, beyond nutritional stimulation by employing a diet restrictive strategy. CKD and sham mice were given GH (10 mg/kg/day, intraperitoneal) or vehicle for 42 days. Each mouse was individually housed during the study period. CKD mice treated with vehicle were fed ad libitum. We measured caloric intake in vehicle-treated CKD mice by multiplying total mouse 5015 diet consumption during the 42 days (in grams) with the metabolizable energy value of the diet (3.59 kcal/gram). The average daily energy intake for vehicle-treated CKD mice was calculated and expressed as kcal/mouse/day. We then fed the same amount of mouse 5015 diet based on the recorded average daily energy intake for vehicle-treated CKD mice to other groups of mice, i.e., CKD mouse treated with GH (10 mg/kg/day, intraperitoneal) as well as sham mouse treated with GH (10 mg/kg/day, intraperitoneal) or vehicle. We fed the mouse daily during the daytime (0900-1200). We measured weekly weight change for each mouse. The schematic study plan for these two studies is illustrated in Figure 1 and Figure 3, respectively.
- Blood chemistry: Specify “standard methods”; Typo in Pierce BCA assay kit; Protein: Specify antibodies and methods used for ATP measurement.
Changes are shown as the followings;
4.3. Serum and Blood Chemistry
Mice were sacrificed and serum samples were collected within 4 hours after the last rhGH or vehicle injection. VetScanâ Comprehensive Diagnostic Profile reagent rotor and the VetScan Chemistry Analyzer (Union City, CA, USA) was used for quantitative determination of BUN and serum creatine concentration (Supplemental Table 1). Concentrations of serum GH in mice were analyzed using commercially available ELISA kits according to the manufacturer’s protocols (Supplemental Table 1).
4.4. Protein Assay for Muscle and Adipose Tissue
A portion of the right gastrocnemius muscle, inguinal WAT and interscapsular BAT was processed in a tissue homogenizer (Omni International, Kennesaw, GA, USA). Protein concentration of tissue homogenate was assayed using a Pierce BCA Protein Assay Kit (Thermo Scientific, catalog 23227, Waltham, MA, USA). Uncoupling (UCP) protein content in muscle and adipose tissue homogenates were assayed. In addition, adenosine triphosphate (ATP) content of tissue homogenate was assessed by using an ATP Assay Kit which relies on the phosphorylation of glycerol that could be quantified by colorimetric or fluorometric methods (Abcam, catalog ab83355, Cambridge, UK). Protein concentration of phospho-JAK2 and total JAK2 as well as phospho-STAT5 and total STAT5 in muscle homogenates was measured (Supplemental Table 1).
- Discussion
- “We also demonstrated that the beneficial metabolic benefits of GH go beyond appetite stimulation.” This conclusion is only correct when looking at the data of Fig1, all molecular analyses were performed in the non-pairfed mice, so might be secondary to improved energy intake.
All molecular and protein studies are performed from those pair-feeding mice. To avoid confusion and misunderstanding, we have highlighted in the revised manuscript.
2.3. GH improves skeletal muscle and adipose tissue energy homeostasis in CKD mice
For the rest of the investigation, gastrocnemius, WAT and BAT tissue from diet-restrictive study were used. We studied the effects of GH on skeletal muscle and adipose tissue energy homeostasis in CKD mice. Protein content of UCPs in gastrocnemius as well as in WAT and BAT was significantly higher in vehicle-treated CKD mice (Figure 4, A, C & E). Inversely, ATP content in gastrocnemius, WAT and BAT was significantly lower in vehicle-treated CKD mice (Figure 4, B, D & F). GH decreased UCPs but increased ATP content in muscle and adipose tissue in CKD mice.
- “Finally, using RNAseq analysis, we assessed the muscle transcriptome” this statement is not correct as it was done in a previous study
We apology for this misleading statement. We have pointed out this issue as followings;
2.7. Molecular mechanism of GH on muscle function by RNAseq Analysis
We previously performed transcriptomic profiling of muscle wasting in CKD by RNAseq analysis and identified 12 differentially expressed genes in muscle [23]. Perturbations of these 12 muscle genes are correlated with impaired muscle and neuron regeneration, enhanced muscle thermogenesis and fibrosis. Hence, we studied the effects of GH on expression of these 12 muscle genes in CKD mice. Notable, GH normalized or attenuated seven out of those 12 differentially expressed muscle genes identified in CKD mice while expression of five muscle genes remained different in GH-treated CKD mice (Figure 8, A-L).
In the discussion, ‘Previously, we have performed RNAseq analysis in gastrocnemius muscle between CKD and control mice and identified the top 12 differentially expressed genes that have been associated with energy metabolism, skeletal and muscular system development and function, nervous system development and function as well as organismal injury and abnormalities [23]. We evaluated the effects of GH treatment on muscle transcriptome in this study. Seven of the 12 muscle genes with variable expression in CKD mice were normalized or reduced by GH (Figure 8). These seven muscle genes—Atp2a2, Cyfip2, Fhl1, Tnnc1, Atf3, Fos, and Itpr1—had aberrant expression patterns that have been linked to enhanced tissue thermogenesis, compromised mechanical muscle properties, poor muscle regeneration, and diminished muscle-neuron regeneration capacity [23].
- “thermogenesis” in muscle, assuming this based on UCP3 expression is incorrect. UCP3 is expressed in low levels, and does not have the same role as UCP1 (for instance, see PMID: 31133866). So this is incorrect wording, in addition to unclear functional relevance. Many functions of UCP3 are known which could contribute to muscle dysfunction in cachexia, and could be discussed instead.
Additional information for relevance of Ucp3 and Ucp1 has been incorporated, page 9, ‘By increasing muscle mass and improving energy efficiency, GH may improve muscle strength. Anaerobic and aerobic energy sources comprise the continuum of energy needed to fuel muscular function. Anaerobic energy systems are stimulated by GH, which suppresses the aerobic energy system. This increases muscle strength. After six months of GH therapy, healthy males showed a considerable improvement in their lower body muscle strength [5]. By "uncoupling" ATP synthesis, UCPs regulate energy homeostasis by dissipating the mitochondrial proton gradient for ATP synthesis and producing heat [25,26]. UCP3 is expressed in skeletal muscle and upregulation of UCP3 has been reported in various conditions characterized by skeletal muscle atrophy, including denervation, diabetes, cancer and sepsis [27]. Gastrocnemius UCP3 protein content along with ATP content was normalized in GH-treated CKD mice (Figure 4, A & B). Putative functions of UCP3 are controversial. Interesting evidence for and against UCP3 involvement in thermogenesis has been published [27,28]. Furthermore, increased muscle expression of UCP3 has been postulated to modulate oxidative stress and lipotoxicity in rat model of cachexic sepsis [29]. GH therapy reduced abnormal UCP1 and ATP content in WAT and BAT in CKD mice (Figure 4, C-F). However, the precise role of UCP1 in disease-associated cachexia in human is still a topic of debate. Several studies have described UCP1 expression, as a biomarker of WAT browning, a critical component of WAT dysfunction in cancer cachexia [17,18,30]. Results also suggested that an UCP-1 independent cascade could also regulate adipocyte homeostasis and influence tumor-induced WAT wasting [31]. Moreover, activation of BAT has been associated with hypermetabolism in cachexia but information from human study is scarce. A recent study investigated the relationship between activation of BAT and hypermetabolism in patients with emphysematous COPD (chronic obstructive pulmonary disease). BAT activity and gene expression of beige markers of BAT in WAT (Tmem26, Cidea, CD137, Shox2 and Tnfrsf9) were not different between COPD patients versus controls [32]. Medications may influence the sympathetic nerve system and BAT metabolism. Adrenergic receptor blockers and calcium channel blockers have been commonly used by COPD patients. Recent studies suggested that possible involvement of b-adrenergic receptor in BAT metabolism in humans and rodents [33,34]. Data also indicated that calcium channel blockers regulated adipogenesis and BAT browning [35,36]’.
Comments from reviewer 2
I appreciated the clarity of the manuscript and found the results of GH treatment in CDK related cachexia quite convincing. GH is restoring the expression of several relevant genes that have been previously characterized to be involved in the cachectic process both for adipose tissue and skeletal muscle. Importantly, effects were independent from food intake, and GH did not alter much the expression of genes in the sham group.
We appreciate the positive comments.
- Line 21 : Myd88 and not Myf88; All figures titles : I would remove “shows that” from every figure legend; Figure 2 : I think the authors wanted to use interscapular instead of intercapsular; Line 152 : “intramuscular WAT in the gastrocnemius muscle “. Not clear. Figures title only refer to inguinal WAT, interscapular BAT or gastrocnemius
We have corrected those issues.
- Line 153 to 157 : Not necessary , statistical representation on the graphs are sufficient to understand what has been compared to what; Lines 177 to 186 : Each single figure is not referred in the text eg : Fig 4A , Fig 4 B etc; Line 210 : results are analyzed and expressed as in figure 2; Line 321 : Total STAT5 and not STAST5
We have corrected those issues.
- The shapes and colors chose for sham and CKD groups in graphs are extremely similar, maybe more contrasting shapes or colors would help to distinguish better the two groups.
We have changed the shapes and color for sham and CKD mice.
- I noticed that in most of the qPCR data, the value for the control group sham + vehicle is not strictly equal to 1. What are you using to calibrate your experiment, a specific sample or the totality of you control group ?
For gene expression, we have used totality of each mouse data point in the same group.

Reviewer 2 Report
The work presented here evaluates the effects of Growth Hormone (GH) treatment in chronic kidney disease- associated cachexia. The authors first evaluate two different doses of growth hormone in an experiment where mice were fed ad libitum. Given that GH is promoting food intake, they performed a second experiment of pair feeding. This strategy allows to distinguish direct effects of growth hormone on cachexia from indirect effect triggered by increased food intake. The authors then describe positive effects of GH on fat mass, lean mass but also muscle functionality. They also show that GH effect on cachexia are mediated by an enhancement of energy metabolism both in muscle and adipose tissue, a decrease in the browning transcriptional signature, a decrease in atrogene expression and other negative regulator of muscle mass and an increase in some pro-myogenic factors. Interestingly they also found that GH was modulating 7 out of 12 genes previously identified as top differentially expressed genes in the skeletal muscle of CDK cachexia.
I appreciated the clarity of the manuscript and found the results of GH treatment in CDK related cachexia quite convincing. GH is restoring the expression of several relevant genes that have been previously characterized to be involved in the cachectic process both for adipose tissue and skeletal muscle. Importantly, effects were independent from food intake, and GH did not alter much the expression of genes in the sham group.
A few comments :
Line 21 : Myd88 and not Myf88
All figures titles : I would remove “shows that” from every figure legend.
Figure 2 : I think the authors wanted to use interscapular instead of intercapsular.
Line 152 : “intramuscular WAT in the gastrocnemius muscle “. Not clear. Figures title only refer to inguinal WAT, interscapular BAT or gastrocnemius.
Line 153 to 157 : Not necessary , statistical representation on the graphs are sufficient to understand what has been compared to what.
Lines 177 to 186 : Each single figure is not referred in the text eg : Fig 4A , Fig 4 B etc
Line 210 : results are analyzed and expressed as in figure 2
Line 321 : Total STAT5 and not STAST5
The shapes and colors choosed for sham and CKD groups in graphs are extremely similar, maybe more contrasting shapes or colors would help to distinguish better the two groups.
I noticed that in most of the qPCR data, the value for the control group sham + vehicle is not strictly equal to 1. What are you using to calibrate your experiment, a specific sample or the totality of you control group ?
Author Response
Comments from reviewer 2
I appreciated the clarity of the manuscript and found the results of GH treatment in CDK related cachexia quite convincing. GH is restoring the expression of several relevant genes that have been previously characterized to be involved in the cachectic process both for adipose tissue and skeletal muscle. Importantly, effects were independent from food intake, and GH did not alter much the expression of genes in the sham group.
We appreciate the positive comments.
- Line 21 : Myd88 and not Myf88; All figures titles : I would remove “shows that” from every figure legend; Figure 2 : I think the authors wanted to use interscapular instead of intercapsular; Line 152 : “intramuscular WAT in the gastrocnemius muscle “. Not clear. Figures title only refer to inguinal WAT, interscapular BAT or gastrocnemius
We have corrected those issues.
- Line 153 to 157 : Not necessary , statistical representation on the graphs are sufficient to understand what has been compared to what; Lines 177 to 186 : Each single figure is not referred in the text eg : Fig 4A , Fig 4 B etc; Line 210 : results are analyzed and expressed as in figure 2; Line 321 : Total STAT5 and not STAST5
We have corrected those issues.
- The shapes and colors chose for sham and CKD groups in graphs are extremely similar, maybe more contrasting shapes or colors would help to distinguish better the two groups.
We have changed the shapes and color for sham and CKD mice.
- I noticed that in most of the qPCR data, the value for the control group sham + vehicle is not strictly equal to 1. What are you using to calibrate your experiment, a specific sample or the totality of you control group ?
For gene expression, we have used totality of each mouse data point in the same group.
Round 2
Reviewer 1 Report
The authors have addressed the majority of my concerns and have significantly improved the manuscript.